 **eLife**

# Minimal-assumption inference from population-genomic data

Daniel B Weissman[1,2]*, Oskar Hallatschek[2]*

[1]Department of Physics, Emory University, Atlanta, United States; [2]Department of Physics and Integrative Biology, University of California, Berkeley, Berkeley, United States

**Abstract** Samples of multiple complete genome sequences contain vast amounts of information about the evolutionary history of populations, much of it in the associations among polymorphisms at different loci. We introduce a method, Minimal-Assumption Genomic Inference of Coalescence (MAGIC), that reconstructs key features of the evolutionary history, including the distribution of coalescence times, by integrating information across genomic length scales without using an explicit model of coalescence or recombination, allowing it to analyze arbitrarily large samples without phasing while making no assumptions about ancestral structure, linked selection, or gene conversion. Using simulated data, we show that the performance of MAGIC is comparable to that of PSMC' even on single diploid samples generated with standard coalescent and recombination models. Applying MAGIC to a sample of human genomes reveals evidence of non-demographic factors driving coalescence.

## Introduction

The continuing progress in genetic sequencing technology is enabling the collection of vast amounts of data on the genomic diversity of populations. These data are potentially our richest source of new information on evolutionary history. The challenge now is to figure out how to extract this information – how to learn as much as possible about the history of populations from modern data sets of many densely-sequenced individuals.

Perhaps the best-established approach to historical inference from genetic data is to fit demographic models to the site frequency spectrum (SFS) (e.g., *Gutenkunst et al. (2009)*; *Excoffier et al. (2013)*; *Liu and Fu (2015)*). The SFS is easy to calculate, even from very large samples, and demographic models can be fit to it without a specific model of recombination, but it neglects all information about how diversity is distributed across the genome, treating each site independently. This is a natural approximation for samples sequenced only at a sparse set of weakly-linked loci, but in large whole-genome samples much of the information is contained in associations among different polymorphisms. Because SFS-based approaches cannot use this information, the number of model parameters they can reliably estimate is limited by the sample size, regardless of how much of the genome is sequenced (*Myers et al., 2008*; *Bhaskar and Song, 2014*).

Recently, an alternative approach has been developed in which a hidden Markov model is used to explicitly model recombination along the genome (the 'sequential Markovian coalescent', SMC or SMC', *McVean and Cardin (2005)*; *Marjoram and Wall (2006)*; *Paul et al. (2011)*), vastly increasing the amount of information that can be gleaned from samples of a small number of individuals (*Hobolth et al., 2007*; *Li and Durbin, 2011*; *Harris and Nielsen, 2013*; *Sheehan et al., 2013*; *Schiffels and Durbin, 2014*; *Steinrücken et al., 2015*). But this requires modeling coalescence and recombination throughout the analysis, and as a result becomes computationally intractable for large samples (while still being far less computationally intensive than modeling the full ancestral

*For correspondence:
dbweissman@gmail.com (DBW);
ohallats@berkeley.edu (OH)

**Competing interests:** The authors declare that no competing interests exist.

recombination graph). Additionally, for an increasing number of populations, we have multiple genomic sequences but know almost nothing about their natural histories, including plausible historical demographies and patterns of recombination and selection; this is true even for some model organisms (see *Alfred and Baldwin (2015)* and other articles in series). It is generally unclear how deviations from the underlying models, e.g., past population structure or gene conversion, affect the inferences of these methods.

Here we present a method for Minimal-Assumption Genomic Inference of Coalescence (MAGIC) that infers the patterns of ancestry and recombination in an arbitrarily large sample of genomes while making only minimal, generic assumptions about recombination, selection, and demography. MAGIC finds approximate distributions of times to different common ancestors of the sample. These distributions can then be used to fit and test potential models for the history of the population, including the simplest model of a single time-dependent 'effective population size', $N_e(t)$. MAGIC strikes a balance between the SFS- and SMC-based approaches, using the distribution of diversity across genomic windows of varying size to generate a description of the single-locus coalescent process that contains far more information than the simple SFS without using a detailed model for recombination.

## Results

### Approach

The key fact underlying MAGIC is that the relationship between the population parameters (such as recombination rates, historical demography, and selection) and genomic data is entirely mediated by the coalescent history of the sample (*Figure 1a*). We therefore take it as our goal to learn the coalescence time distribution directly from the data without needing a model for the population dynamics. Once one knows the coalescent history, the genomic data typically contains no additional information about the population parameters (unless, e.g., the selective values of different alleles are known), and one can fit or evaluate a wide range of models without having to re-analyze the full data set every time. In simple cases, such as when the population is described by a single effective population size, $N_e(t)$, this can be done analytically (*Gattepaille et al., 2016*), while in general it can be done via Approximate Bayesian Computation (ABC).

Essentially, MAGIC uses the variability in the density of polymorphisms across a wide range of length scales to learn the genome-wide distribution of coalescent histories. This technique is inspired by *Li and Durbin (2011)*'s method, PSMC, and its successor MSMC (*Schiffels and Durbin, 2014*), which use the fact that SNPs tend to be dense in regions with a long time to the most recent common ancestor (TMRCA), and sparse in regions with short TMRCAs (*Figure 1*, top left and middle left). Thus, the distribution of SNPs across the genome can be used to infer the distribution of *local* coalescence times. But while PSMC and MSMC use models for coalescence and recombination to assign a coalescence time to each locus, MAGIC estimates the genomic distribution of times directly, bypassing the need for explicit modeling. To do this, MAGIC first splits the genome into windows and finds the distribution of genetic diversity across windows, that is, the histogram of the number of polymorphic sites per window of a given length (*Figure 1*, bottom left; Figure 6, top left). (This can be seen as a simplified version of the 'blockwise counts of SFS types (bSFS)' introduced by (*Bunnefeld et al., 2015*).) This histogram is then used to estimate the distribution of *window-averaged* coalescence times (*Figure 1*, bottom right; Figure 6, bottom). For small windows, these times are essentially the true single-locus coalescence times, but the inference is noisy due to the small number of mutations in each window. For large windows, the inference of the window-averaged distribution is more precise, but this distribution is far from the single-locus distribution because windows typically span multiple segments with different coalescence histories. The basic trick of MAGIC is that rather than choosing one window length, it integrates the information gathered from a wide range of different window lengths to find the small-length limit – the true single-locus distribution (*Figure 1*, center right).

MAGIC's accuracy is comparable to the state-of-the-art model-driven method MSMC (*Schiffels and Durbin, 2014*) on small data sets that conform to MSMC's assumptions (see 'Single diploid samples' in 'Results'). More importantly, it can also analyze larger samples, and can be useful in analyzing data from populations with features (such as gene conversion, ancestral structure, or

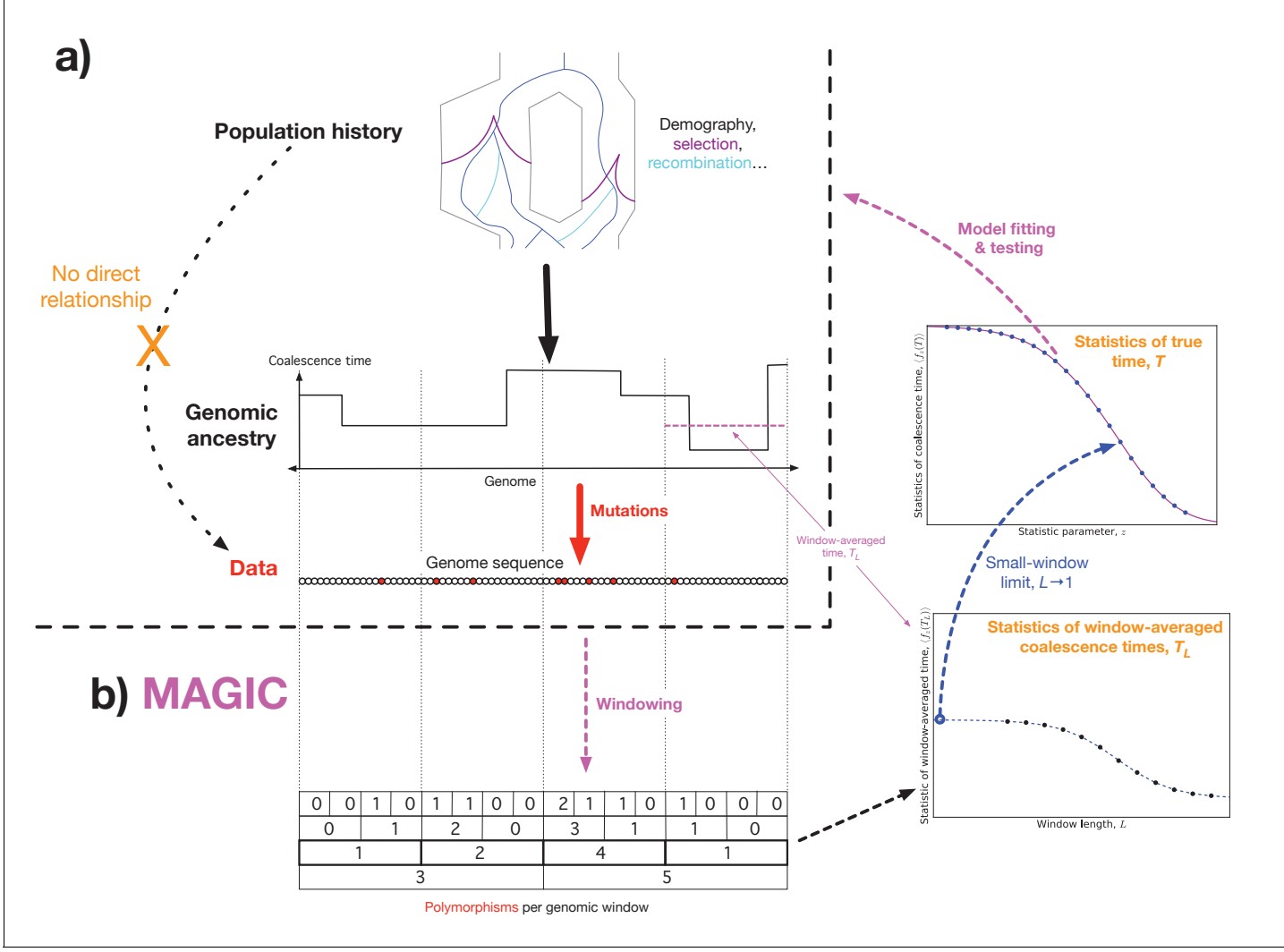

**Figure 1.** Outline of approach. (**a**) Concept: we would like to infer the history of the population (top) from the sequence data (bottom), but the causal connection between the two is entirely mediated by the coalescent history of the sample (middle). This suggests that it should be possible to extract much of the coalescent information from the data without making strong assumptions about the population dynamics. (**b**) Schematic algorithm: MAGIC first splits the sample into small windows and counts the polymorphisms within each window, then progressively merges pairs of adjacent windows together (bottom left). For each window length $L$, the histogram of window diversity is used to estimate parameters of the distribution of the window-averaged coalescence time $T_L$ (bottom right). Taking the limit of these as $L$ goes to 1 gives the parameters of the distribution of the true coalescence time $T$ (top right), which are then used to fit and test models for the underlying dynamics.

linked selection) that violate MSMC's assumptions, either as a stand-alone inference method or as a way of testing models inferred by other methods that include these features. MAGIC's algorithm, described in 'Methods' below, is designed to be as simple and modular as possible, allowing one to incorporate additional assumptions in situations where more information is available. This also enables the inference of a wide range of parameters, including the distribution of map lengths of blocks of identity-by-descent across the genome (*Ralph and Coop, 2013*). Finally, MAGIC can use the dependence of the window-averaged distributions on window size to learn about the rate of recombination and the variation in recombination rate and the coalescent process across the genome. The variation in the coalescent process is particularly interesting because it is a signature of the effects of non-demographic forces.

## Representing coalescence time distributions

For single diploid samples, the coalescent history is completely described by a single time at each locus. Thus the pairwise coalescence time distribution could equivalently be described by the hazard function (the pairwise coalescence rate, as in *Schiffels and Durbin (2014)*) or its reciprocal (the 'effective population size' $N_e(t)$, as in *Li and Durbin (2011)*). However, when estimating the distribution from noisy data, the procedure that minimizes the error for one description will not in general minimize the error for the others. We focus on estimating the coalescence time distribution itself (rather than its hazard function or $N_e(t)$), primarily because it naturally generalizes to arbitrary sets of coalescent tree branch lengths for larger samples (see below). This also lets us emphasize that the idea that the full coalescent can be described by a single $N_e(t)$ is a model that can be tested (as in Figure 3 and the right panel of Figure 4). For plotting results from analyzing simulations (*Figure 2* and *Figure 3*), we show cumulative distributions rather than densities so that we can plot the actual coalescent histories of samples (black curves in *Figure 2* and left panels of *Figure 3*), which consist of discrete sets of events, and also because the density estimates are very poorly constrained by the data (*Ralph and Coop, 2013*). For the analysis of human data, we plot the more familiar $N_e(t)$ (*Figure 4*, left panel).

We must also choose how to summarize coalescence time distributions within MAGIC's analysis (*Figure 1*, right-hand side). We will use the *Laplace transform* of the density, $\tilde{p}_\tau(z)$. For a formal definition of this quantity and how we estimate it, see the section 'Laplace transforms' of the Methods below. Intuitively, $\tilde{p}_\tau(z)$ represents what the homozygosity would be if the mutation rate had been larger by a factor of $z$, e.g., $\tilde{p}_\tau(1)$ is the observed homozygosity, and $\tilde{p}_\tau(2)$ is what it would have been if the mutation rate had been twice as high (*Lohse et al., 2011*). This means that $\tilde{p}_\tau(z)$ very roughly corresponds to the probability of coalescing within $\sim 1/(z\mu)$ generations, where $\mu$ is the mutation rate, that is, large $z$ tell us about the recent past while small $z$ tell us about the distant past. The mean coalescence time corresponds to $z \sim 1/\hat{\pi}$, where $\hat{\pi}$ is the observed heterozygosity of the sample. We choose this statistic both because of this natural interpretation and because it can be obtained from the observed diversity distribution directly with only very weak assumptions, allowing us to delay introducing stronger assumptions until the very last step of the analysis, when we invert the transform to find the full coalescence time distribution (see Materials and methods below).

## Single diploid samples

To validate our approach, we have tested MAGIC on single diploid samples generated under a range of coalescent models simulated with ms (*Hudson, 2002*). MAGIC accurately infers the distribution of coalescent times from samples with map length and polymorphism density similar to that of a human genome (*Figure 2*, top two rows, solid curves; see Figure 9 and Methods for detailed parameters and additional tests). MAGIC performs nearly as well as MSMC (Kolmogorov-Smirnov distance to the true distribution of $5 - 11\%$ for the simulations shown, compared to $4 - 11\%$ for MSMC). Both methods tend to smooth out sharp transitions in the coalescence distribution as a consequence of regularization. The distribution of map lengths of blocks of identity by descent (IBD) can be inferred with very high accuracy (*Figure 2*, top two rows, dotted curves), improving on MSMC, which sometimes overestimates the amount of very deep coalescence, and correspondingly erroneously estimates a large number of very short blocks. The part of the block-length distribution estimated by MAGIC is complementary to the very long blocks that can be observed directly (as in, e.g., *Ralph and Coop (2013)*). MAGIC is also accurate on genomes simulated with ms under a model in which recombination is dominated by gene conversion (*Figure 2*, bottom row); this can be seen as loosely corresponding to a primarily asexual population, with gene conversion representing homologous recombination. In this case, MSMC's recombination model breaks down and MAGIC's inferences are more reliable.

## Larger samples

For samples of more than two haplotypes, the coalescent history at each locus is described by a tree, rather than a single time. The space of possible trees grows very rapidly with the sample size, so that even with long genomes it is impossible to directly estimate the full distribution. Instead, MAGIC infers the distribution of some small set of features of the trees, such as mean pairwise distance and total branch length, chosen either because they are important in and of themselves or

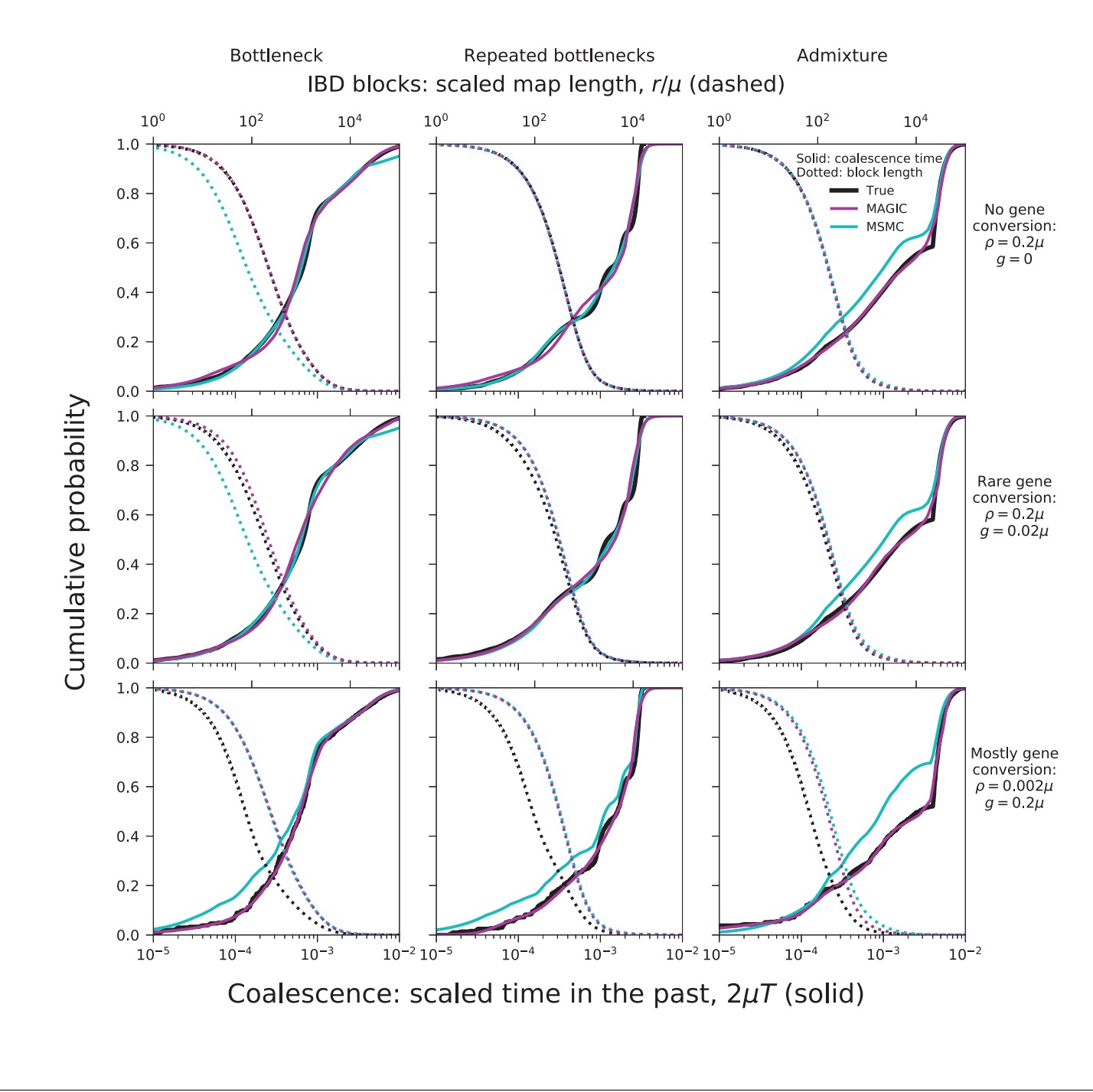

**Figure 2.** MAGIC accurately infers the distribution of coalescence times (solid curves) and lengths of blocks of identity-by-descent (IBD, dotted curves) for pairwise data simulated with ms under several demographic scenarios (*Figure 9*): a single bottleneck (left column), repeated expansions and contractions (middle), and recent admixture of two diverged populations (right). The coalescence time plots show the cumulative distribution, while the IBD block-length plots show the survival function. When crossovers are frequent and gene conversion is rare (top two rows), MAGIC and MSMC are comparably accurate for coalescence times. MAGIC very accurately infers the IBD block length distribution, while MSMC sometimes is inaccurate (e.g., 'Bottleneck' scenario), but otherwise produces a curve nearly indistinguishable from MAGIC's. For frequent gene conversion and rare crossovers (bottom row), the details of the gene conversion process have a strong effect on the IBD block lengths, and neither method can infer their distribution, but MAGIC can still infer the coalescence times. All simulations are of a genome consisting of 100 independent chromosomes, each $10^7$ base pairs long, with per-base mutation rate $\mu$ and present population size $N_0$ such that $N_0\mu = 10^{-3}$. Recombination is via crossovers occurring at rate $\rho$ per base,

*Figure 2 continued on next page*

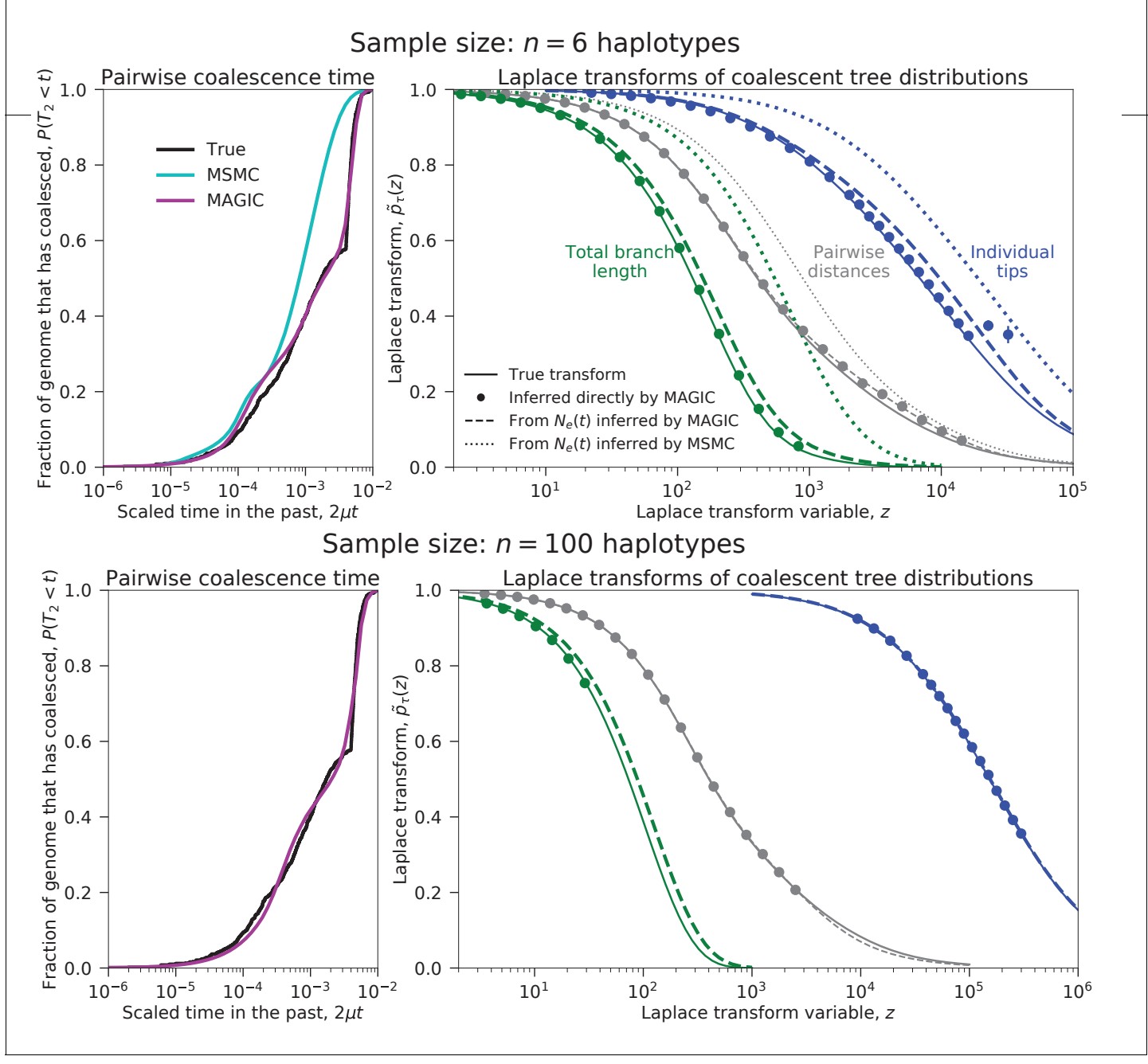

**Figure 3.** Coalescence time distributions and their Laplace transforms for a simulated population with ancestral structure, as inferred from samples of intermediate and large size. Top row: sample size $n = 6$ haplotypes. MAGIC accurately estimates the distribution of the pairwise coalescence time (top left), and the Laplace transforms of the distributions of the total branch length and the lengths of the tips of the branches (top right, green and blue, respectively). The gap between the estimated Laplace transforms (circles) and that obtained by simulating the $N_e(t)$ estimated from the pairwise coalescence time distribution (green and blue dashed curves) suggests that the $N_e(t)$ model is inaccurate. The pairwise Laplace transforms are shown in gray for comparison. MSMC's inferences are less accurate. Bottom row: sample size $n = 100$ haplotypes (too large for MSMC). The gap remains between the estimated Laplace transform of the total branch length distribution and that derived from the pairwise $N_e(t)$, showing that the coalescent process cannot be described by a single $N_e(t)$. However, the tip length distribution matches that predicted from the pairwise comparisons, showing that the $N_e(t)$ model has predictive power for recent times.

because they are sufficient statistics for some model of the coalescent process. For example, MAGIC can fit the basic time-dependent effective population size model by estimating the distribution of pairwise coalescence times, and then check whether the fitted model correctly predicts the

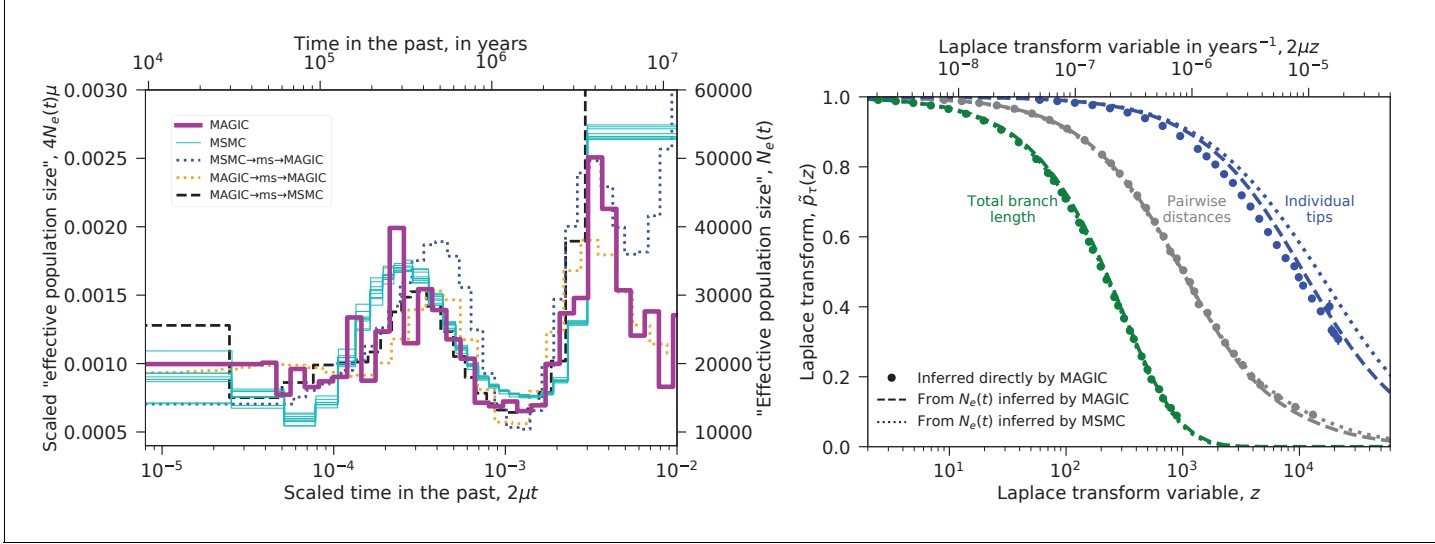

**Figure 4.** Inferred evolutionary history of Yoruban individuals. Left: Inferred 'effective population size' $N_e(t)$ for the nine unrelated Yoruban individuals from the 69 Genomes data set (*Drmanac et al., 2010*). MAGIC's inference from the full sample is similar to MSMC's inference from each individual, differing mostly in the distant past where there is limited data. Dotted lines show inference from samples simulated under the inferred $N_e(t)$'s, indicating that MAGIC's inference in the distant past is likely to be more accurate than MSMC's, but that its inference of the fine-scale structure in $N_e(t)$ is unreliable. Right: Laplace transforms of the distributions of different coalescence times for the same sample. The pairwise coalescence time (corresponding to $N_e$ in the left plot) is in gray, while the total branch length is in green and the distribution of tip lengths is in blue. Points show the values inferred directly by MAGIC, while solid and dashed curves show the results of simulations based on the $N_e(t)$ curves inferred by MAGIC and MSMC, respectively. The pairwise $N_e(t)$ accurately describes the distribution of total branch lengths, but overestimates the tip length Laplace transform (i.e., underestimates the tip lengths). MAGIC's overestimate roughly corresponds to a ∼10% underestimate of the median tip length, while MSMC's corresponds to a ∼30% underestimate, indicating that both underestimate recent $N_e(t)$ from pairwise data.

distributions of other tree features. MAGIC is particularly suited for model-checking, since for all the features beyond the basic pairwise coalescence time one can skip the final, most difficult step in the algorithm – inverting the Laplace transforms – and simply compare the distributions at the level of Laplace transforms.

We test this approach on a sample of six haplotypes from a recently admixed population simulated with ms (demographic parameters in the last column of *Figure 2* and *Figure 9*). MAGIC accurately estimates the distributions of pairwise coalescence times, while MSMC is relatively inaccurate (*Figure 3*, top left). MAGIC also accurately infers the Laplace transforms of the distributions of the total branch lengths and the lengths of the tips of the full coalescent trees (*Figure 3*, top right). Comparing these inferences to the predictions of the $N_e(t)$ models corresponding to the pairwise inferences shows that MAGIC is more accurate than MSMC but is still missing features of the tree distributions, indicating that the model is not a good description of the population history. (The differences appear small in the plot, but are far larger than the uncertainties in the inferences and correspond to substantial differences in the actual distributions.) This failure of the $N_e(t)$ may be why MSMC is inaccurate in this case.

MAGIC's running time on large samples is dominated by the time to read all the data through memory, so it grows only linearly with sample size, meaning that the method can be run on essentially arbitrarily large samples. MAGIC accurately estimates the distribution of pairwise times and the Laplace transforms of the total branch length and tip length distributions in a sample of 100 haplotypes from the same admixed population (*Figure 3*, bottom row). (Curves are not shown for MSMC because it cannot analyze large samples; a sample size of eight from this population caused it to crash.) The total branch length Laplace transform remains different from that predicted by the effective population size model, but the tip Laplace transform matches almost exactly, indicating that the $N_e(t)$ model is accurately describing recent times (post-admixture, $\mu t < 10^{-4}$, roughly corresponding to $z \geq 10^4$) but not ancient times (pre-admixture, $\mu t > 10^{-4}$, roughly corresponding to $z \leq 10^4$), as expected.

In the example above, we have only inferred the distributions (or their Laplace transforms) of pairwise coalescence times, total branch lengths, and tip lengths. But to be able to consider a wide range of models for the population, one must be able to estimate a wide range of parameters. MAGIC can infer the distribution of the total length of any specified set of tree branches. For a given set, MAGIC first filters the polymorphisms in the original data for those that correspond to mutations on the desired branches, and then proceeds with the same analysis as in the basic case of a single diploid sample. For example, to find the distribution of total branch length, MAGIC analyzes the genomic distribution of all polymorphic sites, while to find the distribution of tip lengths, it only looks at singletons on specific haplotypes. For data with no linkage information, the tree features that can be estimated correspond to components of the site-frequency spectrum (SFS), but while the SFS just gives estimates of the *means* of the lengths of different sets of branches, MAGIC infers full distributions.

## Human data

We used MAGIC to analyze the nine diploid sequences from unrelated Yoruban individuals in Complete Genomics' '69 genomes' data set (*Drmanac et al., 2010*). The pairwise coalescence time distribution inferred from the heterozygosity distribution was similar to that obtained with MSMC run on individual samples. These distributions are plotted in the left panel of *Figure 4* as 'effective population sizes' (solid lines), while the underlying Laplace transform estimate is plotted in the right panel. The distributions differ mainly in the distant past where the data is limited. The fact that MSMC finds results from single individuals similar to those that MAGIC finds from the whole sample shows that MSMC is the more powerful method in this case. (MAGIC does not consistently detect the hump in $N_e$ at $2\mu t \sim 3 \times 10^{-4}$ when run on single individuals.)

We re-ran the analysis on samples simulated with ms using the inferred $N_e(t)$ trajectories; the results are shown in the left panel of *Figure 4* as dotted lines. These show that MAGIC's results are more accurate than MSMC's for the distant past: MAGIC recaptures a simulated ancient decrease in $N_e(t)$ (orange) and partially recaptures a large ancient $N_e(t)$ (blue), while MSMC misses the ancient decrease in $N_e(t)$ (black). The fine-scale fluctuations that appear in MAGIC's inferred $N_e$ for $2\mu t \sim 2 \times 10^{-4}$ are somewhat puzzling. The fact that MAGIC reproduces a smooth simulated trajectory (blue) and that MSMC fails to capture simulated fluctuations (black) would seem to suggest that they are a real feature of the data, but the fact that MAGIC also fails to capture simulated fluctuations (orange) argues against this. Instead, the fluctuations may be an artifact caused by some non-demographic feature of the real data that is missing in the simulations, potentially related to biases in sequencing coverage or SNP-calling.

We also inferred the Laplace transforms of the distributions of two other coalescence times: the total branch length, and the lengths of the tips of the coalescent trees (*Figure 4*, right), and compared them to ms simulations of the inferred $N_e(t)$ demographies. The total branch length is close to that predicted by pairwise $N_e(t)$, but the tips of the trees are substantially longer. This is consistent with the fact that the pairwise inferences do not detect the recent Yoruban population growth, that is, underestimate the recent effective population size. (It could also be a signature of false-positive singleton SNPs.) Note that even if we did not know that the Yoruban population had been recently increasing, MAGIC's result for the tips would tell us that the estimates of the left tail of the pairwise coalescence time distribution were inaccurate. To reconcile the results, one could try to find an $N_e(t)$ trajectory that fits all the points in the right panel of *Figure 4*. This could be done via ABC as in ABLE (*Beeravolu Reddy et al., 2016*), simulating the coalescent under different $N_e(t)$ and accepting trajectories that fit better. Unlike in ABLE, all the points are single-locus statistics, so these could be just single-locus simulations with no recombination, greatly reducing the computational requirements. Alternatively, one could simply try another inference method that is better able to resolve recent times (e.g., *Terhorst et al. (2017)*'s SMC++) and then check whether the resulting $N_e(t)$ matches MAGIC's inferences. Both of these approaches could also be generalized to include coalescents that cannot be described by an $N_e(t)$ (e.g., for population structure, one could include it in an *ad hoc* ABC model, or one could use an existing method such as ABLE and test its results).

## Inferring differences among chromosomes: recombination and coalescence

MAGIC's coalescence-time inference is designed to be robust to the form of recombination, but it can also be used to learn about recombination. To do this, rather than simply taking the small-window limit of Laplace transforms of the window-averaged coalescence time, one can look at how they change as a function of window size. In general, besides the small-window limit in which almost all windows lie within IBD blocks, there should also be a long-window limit in which almost all windows contain many IBD blocks. In between these two there is a transitional regime where the window length lies within the bulk of the distribution of IBD block lengths; finding this transitional length gives an estimate of the recombination rate.

Because the transition from the small-window to the large-window limit is not very sharp (*Figure 6*, bottom right), this estimate of the recombination rate is very rough. A more precise estimation requires a specific model of recombination and coalescence, like the one used by MSMC. But even if one does not have a good model for the dynamics of a population, one can make the assumption that all autosomes are experiencing roughly the same dynamics, whatever they are. (This assumption is implicit in all demographic inference from full genomes.) In that case, the dependence of the Laplace transforms of the window-averaged coalescence time on window length should be similar across autosomes, and differences in average recombination rates across chromosomes should be detectable as rescalings of the window lengths. MAGIC can therefore use these rescalings to precisely estimate *relative* recombination rates.

As an example of this approach, we analyze each autosome across the nine unrelated Yoruban individuals in the data set. We find that the Laplace transforms of their window-averaged coalescence time distributions all show a similar dependence on window length (*Figure 5*, top left). Up to a rescaling in length, the autosomes appear very similar (*Figure 5*, top center), with the exception of 19. The collapse of the remaining 21 autosomes suggests that they differ primarily in the amount of very recent coalescence (rescaling the heterozygosity) and in average recombination rates (rescaling the window lengths). The scaling factors for the window lengths therefore are an estimate of relative recombination rates, and are indeed very close to values measured by *Kong et al. (2002)* (*Figure 5*, bottom), with the exception of chromosome 19, as expected.

It is no surprise that chromosome 19 is an outlier in coalescence: it has a much higher gene density than the other chromosomes (*Grimwood et al., 2004*), and is therefore likely to have a much higher fraction of loci under selection and affected by linked selection (*Hernandez et al., 2011*). However, the other autosomes do not have identical gene densities, and there are several large regions with unusual patterns of diversity, such as the MHC locus and flanking regions on chromosome 6. Indeed, even after rescaling, there is still more residual variation in coalescence across the 21 similar autosomes than would be expected by chance under the genomic coalescence time distributions inferred by MAGIC and MSMC (*Figure 5*, top right). This variation must be due to non-demographic factors driving coalescence; the fact that they are readily detectable suggests that one should be cautious in interpreting the details of the results of model-based inference in terms of demography.

## Discussion

The MAGIC algorithm bridges the gap between fast but limited SFS-based approaches to demographic inference and model-based approaches that are limited to small sample sizes, allowing far more information to be extracted from large, high-coverage samples. To see the difference between MAGIC and an SFS-based approach, consider the information that can be gained from sites with singleton polymorphisms. Under an approach that treats sites independently, these can be summarized by one number – their genomic frequency – which can only be used to estimate one number – the mean total tip length of coalescent trees. MAGIC, in contrast, also considers how clustered the sites are over a wide range of lengthscales, allowing it to estimate not just the mean, but the whole distribution of total tip lengths.

While this manuscript was in preparation, two other methods extending the SFS by using linkage information were posted. *Terhorst et al. (2017)*'s SMC++ uses the SFS to augment the pairwise coalescent inference approach of PSMC; *Beeravolu Reddy et al., 2016*'s ABLE uses ms to fit demographic models to *Bunnefeld et al., 2015*'s 'blockwise SFS' statistics. These methods share MAGIC's

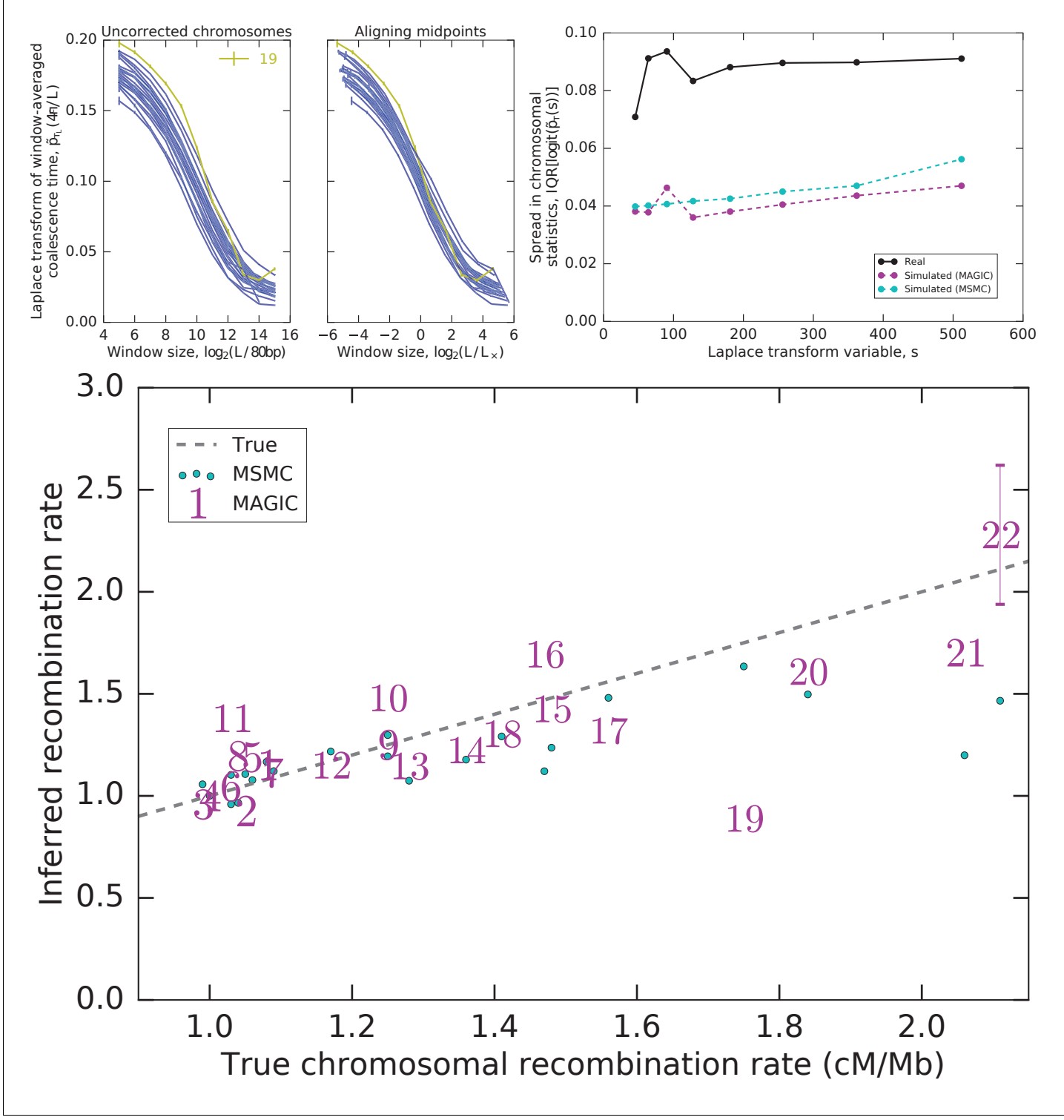

**Figure 5.** Integrating diversity patterns across length scales allows to compare recombination rates among chromosomes and to test whether the observed patterns can be explained by a single shared demographic history. Top left: One value of the Laplace transform of the window-averaged coalescence time distribution as a function of window size for each autosome. The curves for most autosomes appear similar, but shifted both vertically and horizontally. Top center: Much inter-autosomal variation can be explained by variation in recombination rates: the curves are similar under a rescaling of window lengths (a horizontal shift such that the midpoints $L_\times$ of the curves align), except for chromosome 19 (yellow), which appears to have a different pattern of coalescence. Top right: There is substantial variation in the asymptotes that cannot be explained by variation in recombination rates, and is more than expected from intrinsic coalescent stochasticity. Plot shows the interquartile range (IQR) of the Laplace transform

*Figure 5 continued on next page*

*Figure 5 continued*

of the coalescence time distribution across chromosomes for the actual data as well as simulations of the pairwise coalescent histories inferred by MAGIC and MSMC (*Figure 4*). (Note that IQR, unlike variance, is insensitive to outliers.) Bottom: The rescaling of window sizes needed to align the different autosomes gives an estimate of their relative recombination rates which is very close to the values obtained by *Kong et al. (2002)* ('True'). For chromosomes other than 22, the inferred error bars are smaller than the size of the markers.

aim of taking advantage of linkage information for large samples, and as mentioned above, ABLE's initial summary of the genomic diversity is related to MAGIC's. The fundamental difference is that both SMC++ and ABLE use explicit coalescence and recombination processes (SMC' for SMC++, ms's ancestral recombination graph for ABLE) to fit specified demographic models, while MAGIC focuses on just estimating the coalescence time distributions.

MAGIC is designed to be complementary to the existing inference methods, which largely rely on fitting simplified demographic models that neglect selection (*Schraiber and Akey, 2015*). Because MAGIC makes no assumptions about whether coalescence is driven by demography or selection, and only minimal assumptions about mutation and recombination, it can be used as a first-pass analysis of genomes from species whose natural histories are not already well-known, with its results informing the choice of more detailed, model-based methods that use additional information outside of the sample sequences. Conversely, MAGIC can be used for a necessary final step missing in many demographic inference projects: model checking. To evaluate a model produced by other methods, one can use MAGIC to estimate additional parameters beyond those used to fit the model, and then test whether the model reproduces those values. If it does, this can be a crucial sign that the model is capturing important features of the underlying history, while if it does not, the deviations can point to ways in which the model needs to be adjusted (as with the inferences of the recent past of the Yoruban population above).

Finally, MAGIC's estimated Laplace transforms could also be used directly to fit population models (including non-standard ones incorporating, for instance, linked selection, that are not implemented in existing inference methods). Because MAGIC converts the genomic distribution of diversity into the Laplace transforms of single-locus coalescent times, fitting models to its results requires only single-locus coalescent simulations or calculations, which are much less computationally intensive than multi-locus ones. They can thus reasonably be found analytically or via Approximate Bayesian Computation for many models.

Even for populations for which there are good *a priori* models, the minimal-assumption approach has advantages. Because MAGIC has a modular structure and is not tailored to a specific population model, it can be used to quickly analyze many populations with very different dynamics, with each population's model incorporated in just the last step of the analysis. Similarly, for any given population, MAGIC can estimate many different parameters describing coalescence and recombination to answer multiple questions about the historical dynamics. Finally, not using any explicit model of coalescence and recombination keeps MAGIC's algorithm simple enough that it runs quickly even on very large sample sizes, and that users familiar with Python can understand and modify it.

There are a number of potential modifications to MAGIC that users could make. At a minimum, there are likely to be technical improvements to the estimation methods that would allow it to get more information out of the data. More interestingly, the range of parameters estimated by MAGIC could be extended. In particular, MAGIC currently infers the distributions of features of coalescent trees that can be found from unphased, unpolarized polymorphism data, but it could be extended to take advantage of this extra information when available. It would also be possible to extend MAGIC so that it would infer *joint* distributions of different coalescence times, rather than just all the marginal distributions. This would greatly increase the amount of information that could be extracted from extremely large data sets such as are likely to be available in the near future.

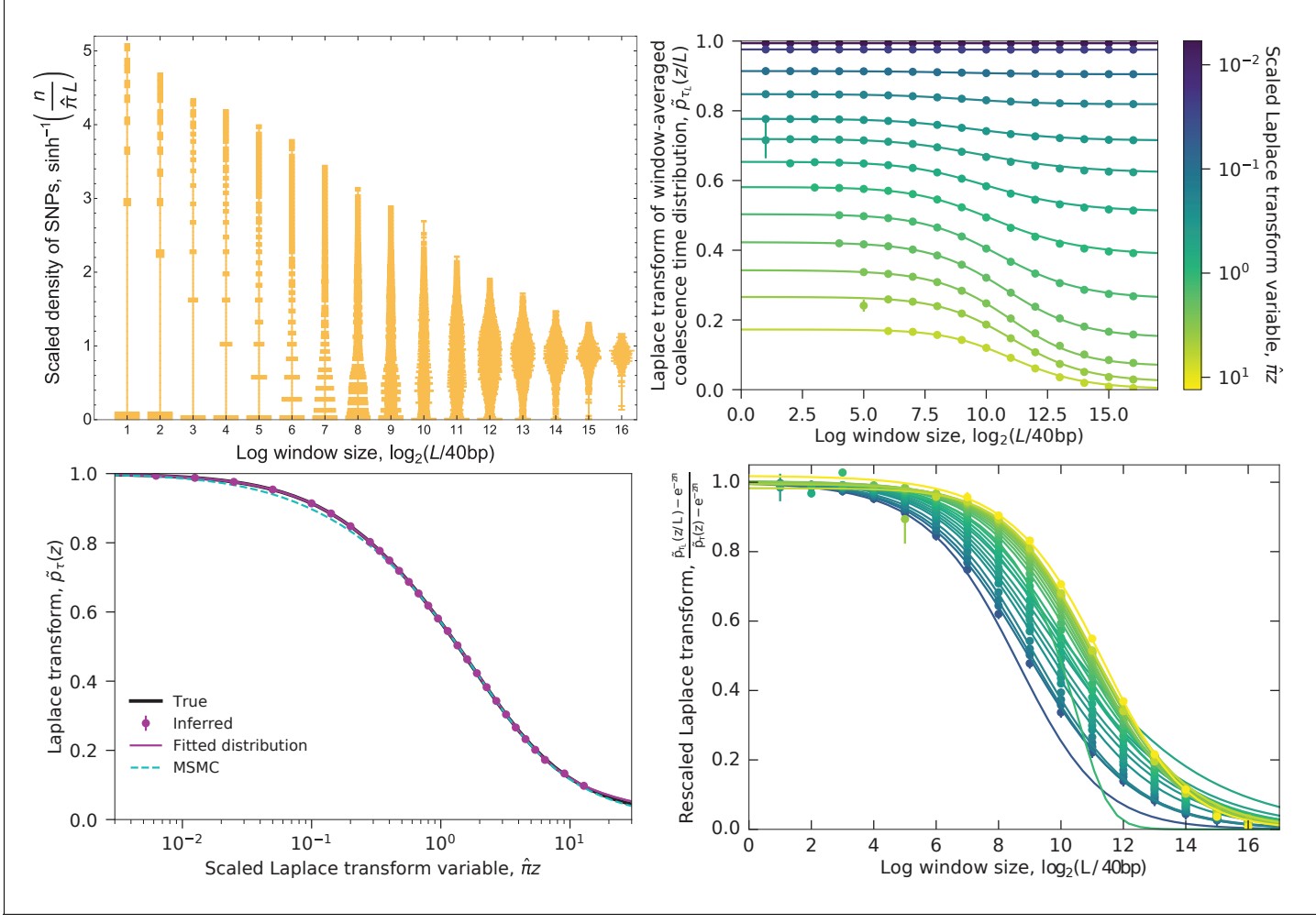

**Figure 6.** Example of converting the distribution of SNPs to the coalescence time distribution. Data are from the simulated 'bottleneck' demography (**Figure 2** and **Figure 9**, left columns). Top left: Distribution of SNP densities across windows of different lengths, normalized by the heterozygosity $\hat{\pi} \approx 1.3 \times 10^{-3}$. The width of a bar represents the fraction of windows with a given SNP density. The densities are shown on an arcsinh scale (approximately logarithmic for large values but linear for small values). At short lengths, the distribution is concentrated at zero, while at long lengths it is bunched near the mean, with the best spread in between. Top right: The Laplace transform $\tilde{p}_{\tau_L}(z/L)$ of the window-averaged coalescence time distribution $p_{\tau_L}$ as a function of window length $L$ and Laplace transform variable $z$. Points show the estimates derived from the SNP density distribution (using **Equation 3**, with error bars given by **Equation 4**), curves are sigmoid fits (**Equation 5**). For small $z$ (long times), small windows can be used to estimate the left asymptote $\tilde{p}_{\tau}(z)$, while for large $z$ (short times), the estimates from small windows diverge (missing points with $L \ll z$) and longer windows need to be used. Typical coalescence times correspond to $\hat{\pi} z \sim 1$ (blue-green). Bottom left: Laplace transform of the coalescence time distribution. Points are the left asymptotes of the $\tilde{p}_{\tau_L}(z/L)$ curves in the top right panel. Black curve shows the true transform, magenta curve shows MAGIC's fitted distributions (the gamma mixture and piecewise exponential forms give indistinguishable curves), cyan curve shows the Laplace transform of MSMC's estimated coalescence time distribution. All the curves are close, but differ slightly for very large $z$, corresponding to very recent times, with MSMC also differing for small $z$ (ancient times). Bottom right: Same as top right, rescaled to show that as $z$ increases, longer windows are sufficient to accurately estimate $\tilde{p}_{\tau}(z)$. But the increase in the minimum sufficient window length is slower than the increase in the minimum length for which $\tilde{p}_{\tau_L}(z/L)$ can be accurately estimated (top right, leftmost points), putting a limit on the maximum $z$ for $\tilde{p}_{\tau}(z)$ can be estimated (bottom left, rightmost point).

# Materials and methods

## Diversity across genomic windows

A sample set of genomes will comprise many blocks of sequence with different coalescent histories; by looking at the distribution of genetic diversity across blocks, one can estimate the coalescence time distribution of the population the sample was drawn from. *Li and Durbin (2011)* and

*Schiffels and Durbin (2014)* try to do this by considering all possible boundaries between the blocks using a hidden Markov model. However, block boundaries are only easy to identify when mutation rates are much larger than recombination rates, which is generally not the case, and describing every possible block becomes impractical for larger sample sizes as the number of blocks proliferates. Instead, we simply divide the genome into windows of a fixed length $L$, and consider the distribution of histories of windows. MAGIC estimates the distribution of a single coalescence time (i.e., coalescent tree parameter) $T$ across genomic positions $x$. For single diploid samples, $T(x)$ is the total branch length (twice the time to the most recent common ancestor at position $x$) and completely characterizes the coalescent history. For larger samples, MAGIC can be used to estimate the distributions of multiple statistics one at a time.

We assume the infinite sites model, in which each mutation occurs at a unique locus. Under this model, the diversity (e.g., heterozygosity in a single diploid sample) in a window of length $L$ starting at position $x_0$ has a distribution that depends only on the *window-averaged coalescence time*, $T_L$ defined as

$$T_L \equiv \frac{1}{L} \sum_{x=x_0}^{x_0+L-1} T(x).$$ (1)

If $L$ is smaller than most block lengths, then windows will typically lie within blocks, and the distribution $P_{T_L}$ of $T_L$ will be close to the distribution $P_T$ of $T$. For very large $L$, each window will average over many blocks, and $P_{T_L}$ will have a narrow support around the mean of $P_T$. Usually, there will be a wide range of intermediate values of $L$ for which windows lie inside long blocks but cover multiple short blocks. Ideally, we would like to use information from windows with lengths throughout this range, preferentially selecting the ones that lie inside long blocks.

Given that $P_{T_L}$ approaches $P_T$, as $L$ decreases, one might be tempted to take $L$ to be as small as possible, but the problem of course is that we cannot see $T_L$ directly; we need to infer it from the number of SNPs in the window. Given $T_L$, and assuming a constant mutation rate $\mu$ per base, the number of SNPs will be approximately Poisson-distributed with mean $\mu L T_L$. (Here we are assuming that $\mu T$ is always small enough that each base has only a small chance of having mutated; this allows us to approximate the underlying sum of binomially-distributed random variables with a Poisson that depends only on $T_L$.) The smaller $L$ is, the lower the signal-to-noise ratio in the number of SNPS will be and the less power we will have to distinguish different values of $T_L$. Thus, we expect that we will get the most information about $P_T$ from an intermediate value of $L$, and should be able to do better still by integrating information from multiple values of $L$.

The total probability that there are $n$ SNPs in a window of length $L$ is the average of the Poisson distribution over all possible values of $T_L$:

$$P_L(n) = \langle e^{-\tau_L} \tau_L^n \rangle_{\tau_L} / n!,$$ (2)

where $\tau_L$ is the window-averaged coalescence time scaled such that it is equal to the expected number of SNPs in the window: $\tau_L \equiv \mu L T_L$. $P_L$ is a Poisson mixture distribution, with mixing distribution given by $P_{\tau_L}$, the cumulative distribution function of $\tau_L$, i.e., the fraction of the genome that is expected to have coalesced by a given scaled time. The observed SNP count distributions $\widehat{P_L}$ thus give us information about the window-averaged coalescence-time distributions $P_{\tau_L}$. We could try to estimate the full distribution $P_{\tau_L}$, but we are primarily interested in the true single-locus coalescence-time distribution $P_\tau$ (where $\tau \equiv \tau_1 \equiv \mu T$). We will therefore focus on estimating just features of $P_{\tau_L}$ that can then be combined to estimate $P_\tau$.

## Laplace transforms

We need to choose which set of parameters describing $P_{\tau_L}$ to estimate. The Laplace transform $\tilde{p}_{\tau_L}(z) \equiv \langle e^{-z\tau_L} \rangle$, evaluated at a set of points $\{z_j\}$, is a natural choice, as it is closely related to the diversity distribution (*Lohse et al., 2011*): *Equation 2* shows that $(-1)^n P_L(n)$ is the $n^{\text{th}}$ Taylor coefficient of $\tilde{p}_{\tau_L}(z)$ about $z = 1$. This has two implications. First, the similarity to the proportion of homozygous windows $P_L(0) = \langle e^{-L\tau_L} \rangle$ means that the Laplace transform has a natural interpretation as an estimate for the proportion of windows of length $L$ that would be homozygous if the mutation rate

were multiplied by $z$. (It is also closely related to the distribution of lengths of IBD blocks – see below.) Second, we can quickly and easily estimate $\tilde{p}_{\tau_L}$ using the plug-in estimator:

$$\widehat{\tilde{p}_{\tau_L}}(z) = \sum_{n=0}^{\infty} \widehat{P_L}(n)(1-z)^n. \tag{3}$$

The estimate lt of the *Equation 3* transform will be accurate for $z$ close to 1, but will blow up for large $z$ – we cannot accurately estimate the amount of very recent coalescence. To make this precise, we need to estimate the error in *Equation 3* due to the stochastic mutation accumulation process. But this depends on the unobserved distribution $p_{\tau_L}$, so we will need a rough estimate of this distribution. We use *Ghosh et al., 1983*'s estimator $\delta_5$ to estimate the value of $\tau_L$ for every window from the observed number of SNPs $n$; roughly, this gives $\widehat{\tau_L}(n) \approx n$, with a correction given by their Equation (2.17) that slightly shrinks the estimated values. (This shrinkage improves the estimate for small values of $n$.) For $n = 0$, where $\delta_5$ would give $\widehat{\tau_L} = 0$, implying that mutations could never occur, we adjust the formula to $\widehat{\tau_L}(0) = \log(2)/K_0$, where $K_0$ is the number of windows with 0 SNPs, to account for the fact that coalescence times may not be exactly 0. We then calculate the standard error that would be introduced by the stochastic mutation accumulation process under $\widehat{P_{\tau_L}}$:

$$\widehat{\sigma^2}\left[\widehat{\tilde{p}_{\tau_L}}(z)\right] = \frac{1}{K} E_{\widehat{P_{\tau_L}}}\left[e^{-z(2-z)\tau_L} - e^{-2z\tau_L}\right], \tag{4}$$

where $K$ is the total number of windows. The accuracy of *Equation 3* and *Equation 4* could be improved by using more sophisticated estimators, but the current ones are the easiest to compute.

## Combining length scales

To combine information from different window lengths, we need to correct for the increase in window-wide mutation rate $\mu L$. We can therefore consider the quantity $\tilde{p}_{\tau_L}(z/L)$ as a function of $L$, holding $z$ fixed, as shown in the bottom panels of *Figure 6*. When $P_{T_L}$ is nearly independent of $L$, this quantity should be nearly constant. (To see this, note that $\tilde{p}_{\tau_L}(z/L) = \langle e^{-z\mu T_L}\rangle$, with no explicit $L$ dependence.) This is the case for very large $L$, when each window averages over many coalescent blocks, and for very small $L$, where each window falls within a coalescent block and $P_{\tau_L}$ approaches $P_\tau$, the distribution we are interested in. We therefore fit a sigmoid curve (specifically, Richards' curve) to $\widehat{\tilde{p}_{\tau_L}}(z/L)$ as a function of $\log(L)$,

$$\tilde{p}_{\tau_L}(z/L) \approx a_z + \frac{b_z - a_z}{[1 + (L/L_\times)^{-c_z}]^{1/\nu_z}}, \text{ with } a_z, b_z \in [0,1]; c_z, d_z, \nu_z > 0; L_\times > 1, \tag{5}$$

and take the left asymptote $a_z$ as an estimate of $\tilde{p}_\tau(z)$. The right asymptote $b_z$ is the long-window limit, and can therefore be estimated directly from the genomic density of SNPs $\widehat{\pi}: b_z = e^{-\widehat{\pi}z}$. ($\widehat{\pi}$ is the heterozygosity in the basic case when $T/2$ is the pairwise TMRCA, but more generally it is the density of SNPs corresponding to the branch statistic being estimated.) To fit the remaining parameters, we use the curve_fit function in SciPy's optimize package on the lowest-$L$ points that have small estimated errors. curve_fit also returns a standard error $\widehat{\sigma^2}(\tilde{p}_\tau(z))$, estimated under the assumption that the errors in $\widehat{\tilde{p}_{\tau_L}}(z/L)$ are independent for different values of $L$. This is obviously not true (all estimates are from the same set of mutations), but even for lengthscales that are only separated by factors of $2$, the correlations in the error appear to be small in simulations, giving final error bars of roughly the right magnitude. While $a_z$ contains the information about the single-locus coalescent, the values of other parameters, particularly $L_\times$, are informative about recombination – see 'Inferring recombination rates' and *Figure 5*.

The sigmoid form sigmoid is flexible enough to fit all of the simulated and real data that we have examined, but we only trust our estimate of $\tilde{p}_\tau(z)$ if the data are close enough to the left asymptote so that the estimate is not very sensitive to the choice of functional form. Effectively, this means that the estimate is close to $\widehat{\tilde{p}_{\tau_L}}(z/L)$ for the smallest $L$ for which the estimated error bars are small, with corrections based on the next few higher lengthscales. But this smallest $L$ depends on $z$, so while for any given value of $z$ only a few lengthscales are important, every lengthscale is important for estimating the Laplace transform for some $z$. Short lengthscales are useful for small $z$ (i.e., long coalescence

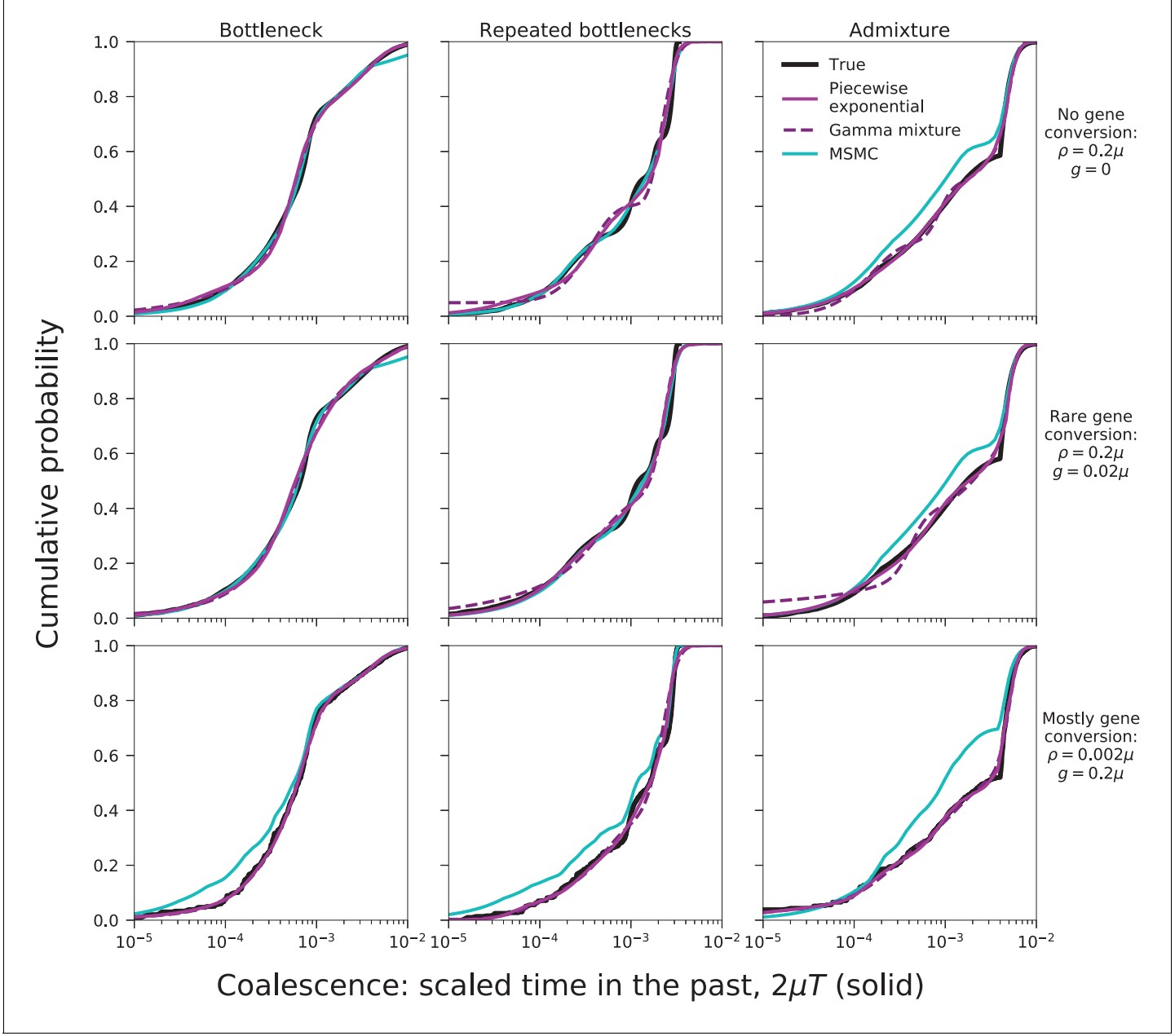

**Figure 7.** Comparison of the functional forms for the coalescence time distribution. The inferred piecewise exponential (solid magenta) and gamma mixture (dashed purple) distributions are very similar, but where they differ, the piecewise exponential form is closer to the true distribution (black). Simulations are the same as shown in *Figure 2*. The two forms' predicted block-length distributions (dotted curves in *Figure 2* are indistinguishable.

times), while long lengthscales are useful for large $z$ (short coalescence times) (*Figure 6*, bottom panels). To see why this is, recall that our estimate $\widehat{p_{\tau_L}}$ is most accurate for arguments near 1, so for each $z$ our most accurate estimate of $\tilde{p}_{\tau_L}(z/L)$ comes from windows of length $L \sim z$. Our smallest window size $L_0$ therefore puts a lower limit on the values of $z$ for which we can estimate $\tilde{p}_{\tau}(z)$, that is, an upper limit on the times we can characterize. Similarly, there is an upper limit on the $z$ values (lower limit on timescale) that we can resolve. This can occur at the value of $z \sim L$ at which the windows become so large that we only have a few per chromosome and no longer have good statistics, or, as in *Figure 6*, at values such that the windows are so long that even the most recently-coalesced ones cover multiple recombination breakpoints, that is, $\tilde{p}_{\tau_L}(1)$ is far from the asymptote $\tilde{p}_\tau(L)$.

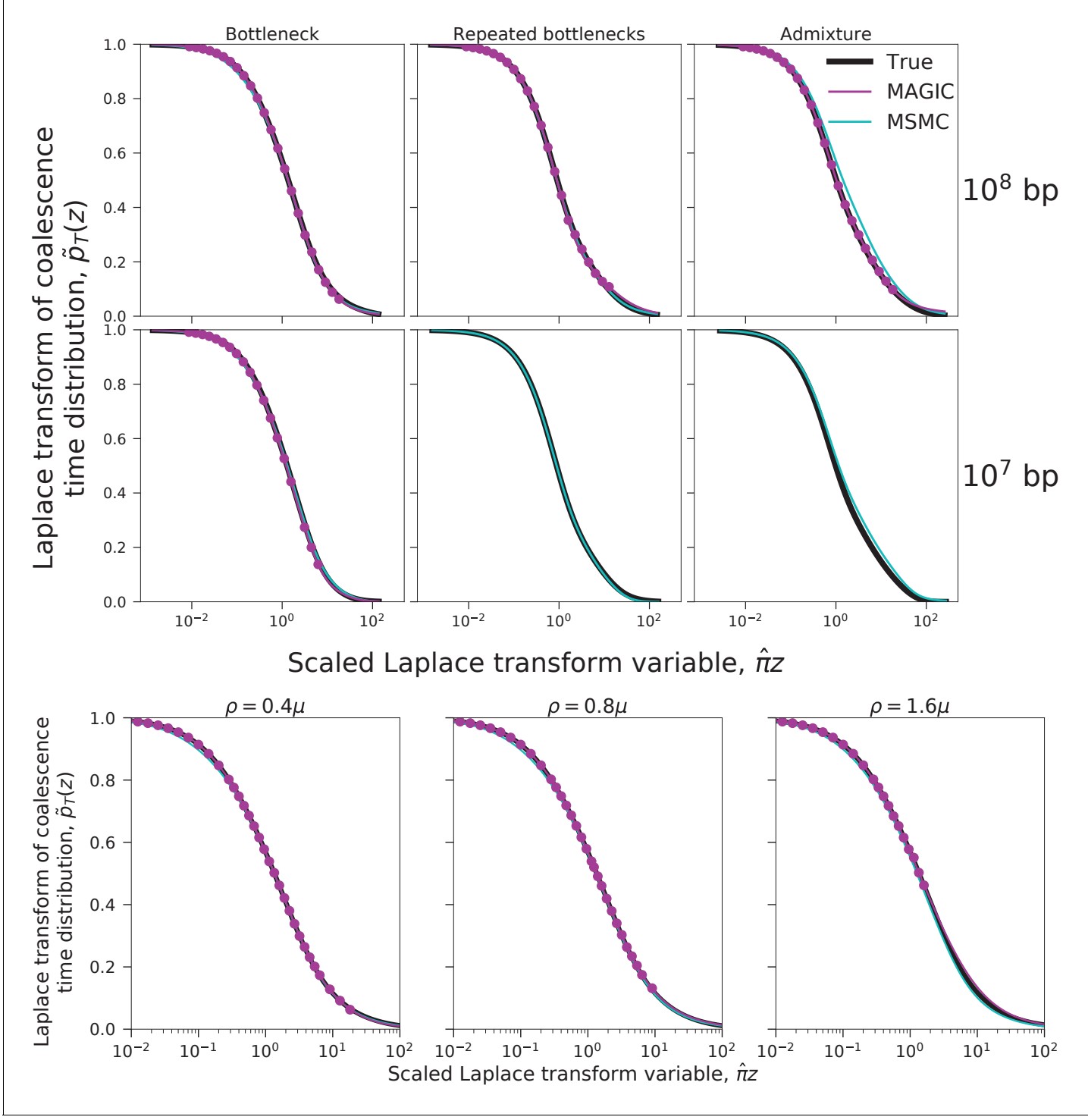

**Figure 8.** Limits on inferring the Laplace transform of simulated pairwise coalescence distributions. Symbols and colors are as in the bottom left panel of *Figure 6*. Top: MAGIC can usually accurately infer the Laplace transform from $10^8$ sequenced bases (top row), but often fails when the data is limited to $10^7$ bases (bottom row, right two panels). Even under limited data, the inferences that it does manage to make remain accurate (bottom row, left panel). Other simulation parameters are as in the top row of *Figure 2*. Bottom: MAGIC can infer most of the Laplace transform when the crossover rate, $\rho$, is less than the mutation rate, $\mu$ (left two panels), but for $\rho > \mu$ (right panel), it becomes limited to large $z$. Simulation parameters are as in the top left panel of *Figure 2*, with $\rho$ increased.

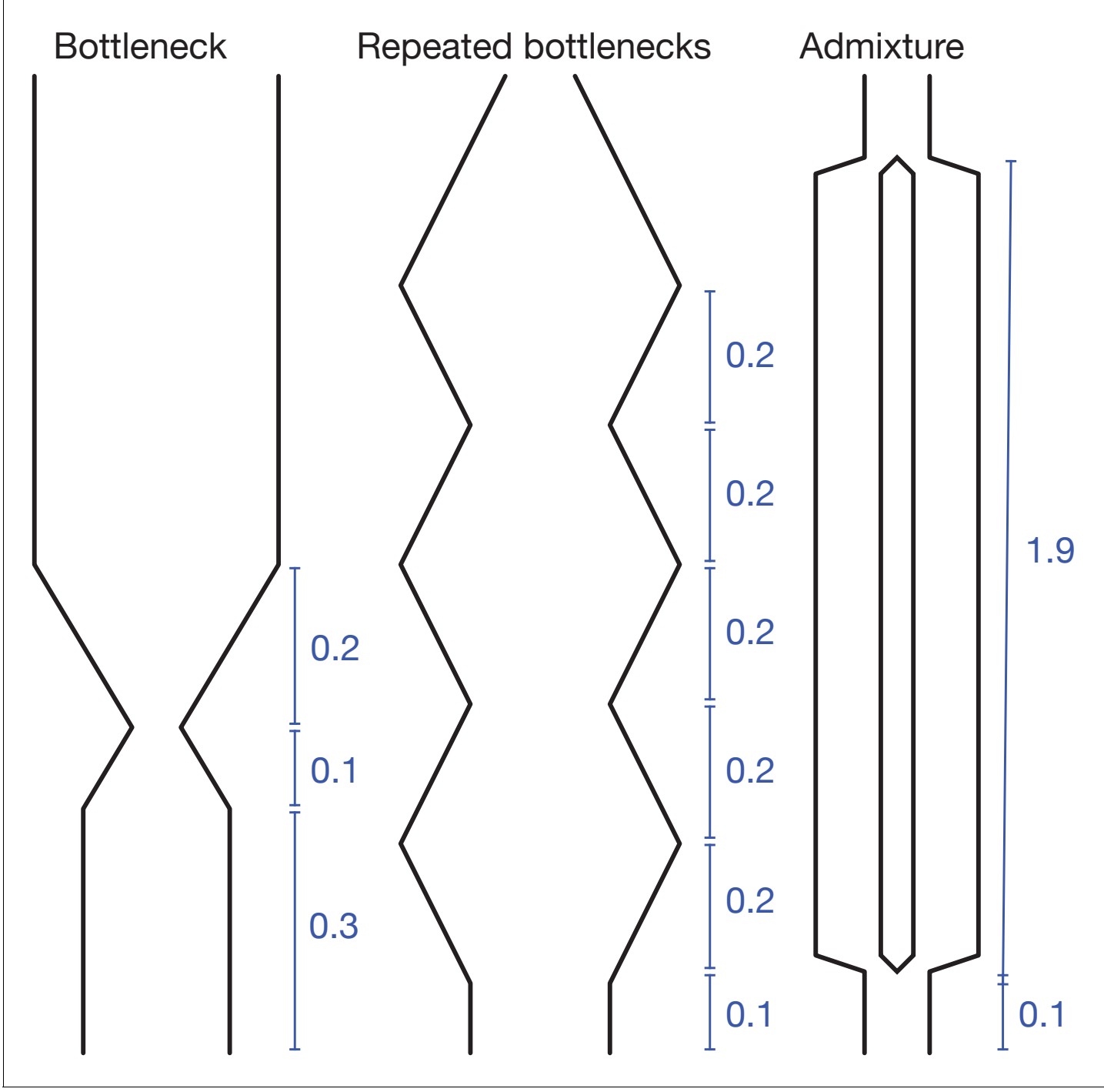

**Figure 9.** Demographic scenarios simulated. All time intervals are in units of $4N_0$. In the 'bottleneck' and 'repeated bottlenecks' scenarios, the population grows and shrinks exponentially at rate $10/(4N_0)$.

## Coalescence-time distributions

Once we have estimates for the Laplace transform of the coalescence time distribution at a set of $\{z_j\}_{j=1,\dots,J}$, we would like to invert the transform to obtain $p_\tau$. The moments of $\tau$ are trivial to find, simply by taking derivatives of $\tilde{p}_\tau(z)$ at $z = 0$, but beyond the first moment these tend to be dominated by rare, deep coalescence events and so are not informative about the bulk of the distribution. Unfortunately, inverting Laplace transforms is a fundamentally hard problem (**Epstein and**

*Schotland, 2008*), and we need to assume some kind of parametric form for $p_\tau$. We have implemented two possibilities in MAGIC.

First, one can assume that $p_\tau$ can be written as a mixture of gamma distributions:

$$p_\tau^{\mathrm{\Gamma M}}(t) = \sum_{i=1}^{\lfloor (J+1)/3 \rfloor} a_i \frac{t^{k_i-1}e^{-t/\theta_i}}{\Gamma(k_i)\theta_i^{k_i}}, \tag{6}$$

where $\sum_i a_i = 1$ and all $a_i$, $k_i$, and $\theta_i$ are positive. This can also be extended to include a possible point mass at $t=0$ when estimating the distribution of features for which it is possible that some trees will have a value of exactly 0. For instance, in a sample of 10 haplotypes, there will typically be some loci for which the coalescent tree has no branches that are ancestral to exactly 5 loci. (More generally, this should be considered when estimating the distribution of lengths of branches that are ancestral to exactly $k$ out of sample of $n$ individuals, for any $n > k > 3$.) Alternatively, one can assume that $p_\tau$ can be written as a piecewise-exponential function, as in MSMC and PSMC:

$$p_\tau^{\mathrm{PE}}(t) = c_{i(t)} \exp\left[ -c_{i(t)}(t - t_{i(t)}) - \sum_{j=0}^{i(t)-1} c_j(t_{j+1} - t_j) \right], \tag{7}$$

where $c_j > 0$, $0 = t_0 < t_1 < \cdots < t_{J-1} < t_J = \infty$, and $i(t) = \max\{i | t_i < t\}$. For pairwise coalescence times, $c_{i(t)}$ has a natural interpretation as the instantaneous coalescence rate, that is, the inverse of $N_e(t)$.

Both of these forms have the computational advantage of having analytic Laplace transforms:

$$\tilde{p}_\tau(z) = \begin{array}{ll} \dfrac{\sum_{i=1}^{\lfloor (J+1)/3 \rfloor} a_i(1+\theta_i z)^{k_i}}{\sum_{i=0}^{J-1} e^{-zt_i - \sum_{j=0}^{i-1} c_j(t_{j+1}-t_j)}\left(1+\frac{z}{c_i}\right)^{-1}\left(1 - e^{-(c_i+z)(t_{i+1}-t_i)}\right)} & \text{(gamma mixture)} \\ & \text{(piecewise exponential),} \end{array} \tag{8}$$

with an additional constant term if a point mass at $t=0$ is included. This means that we can simply fit **Equation 8** to the values $\left\{\widehat{\tilde{p}_\tau}(z_j)\right\}$ without having to deal with inverse Laplace transforms directly. We use the basin-hopping optimization algorithm implemented in SciPy to find the parameters that minimize the scaled squared error $\sum_j \left(\tilde{p}_\tau(z_j) - \widehat{\tilde{p}_\tau(z_j)}\right)^2 / \widehat{\sigma^2}(\tilde{p}_T(z_j))$. For the gamma mixture, we optimize $\{c_i, k_i, \theta_i\}$. For the piecewise exponential, we fix the $\{t_i\}$ to be evenly spread in log space between $1/(2z_J)$ and $1/z_*$ where $\tilde{p}_\tau(z_*) \approx 0.95$ and optimize the rates $\{c_i\}$, with a quadratic penalty on $\log(c_{i+1}/c_i)$ for regularization. These forms are flexible enough to fit all the data that we have tried (see, e.g., **Figure 6**, bottom left – the two forms give indistinguishable curves). The piecewise-exponential form is better-suited for estimating $N_e(t)$ from pairwise coalescence times, and appears to tend to be more accurate in the tails of the distribution (**Figure 7**); we have used it in all curves except the ones in **Figure 7** specifically labeled as showing the gamma mixture form.

## Block-length distributions

Identical-by-descent (IBD) blocks are stretches of the genome that have not undergone recombination since the common ancestor of the block. The distribution of block lengths can be used to infer patterns of relatedness and ancestry (*Li and Durbin, 2011*; *Ralph and Coop, 2013*), but it is hard to measure except for long blocks with very recent common ancestors, because the recombination events are not directly observable. Under the standard assumptions that coalescence is mostly driven by neutral processes (rather than linked selection) and that recombination primarily occurs via crossovers, the distribution of the genetic map lengths of these blocks across the genome is closely related to the Laplace transform of the coalescence time distribution:

$$P(r_{\mathrm{block}} > r) = -\tilde{p}_T'(r)/E[T] = \tilde{p}_T'(r)/\tilde{p}_T'(0). \tag{9}$$

We can estimate the block-length distribution in Morgans simply by approximating the derivative of $\tilde{p}_T$, a much easier problem than inverting the transform. If we want to convert map lengths to numbers of bases, we need to estimate the crossover frequency per base. The dependence of $\tilde{p}_{\tau_L}$ on $L$ (i.e., $L_\times$ in **Equation 5** above) provides a rough estimate; MSMC gives a more precise one. block-length gives the distribution of lengths of blocks that are not interrupted by even 'ghost' recombination events where the coalescent tree does not change ([*Marjoram and Wall, 2006*]'s 'R class' of

events). In the simplest case of a single diploid sample from a well-mixed population evolving neutrally under the Kingman coalescent, these ghost events can be ignored by simply scaling all recombination rates by $2/3$, but in general they must be included, even though they are not directly observable.

## Required sequence length and polymorphism density

For MAGIC to accurately estimate the probability of coalescence within a time interval, the underlying genomic data must contain many regions that coalesced within that interval, and these regions must have enough polymorphisms to put reasonable limits on their coalescence times. For these to be true, there must be at least some minimum length of sequence, and recombination must not be too frequent relative to mutation (or else regions short enough to have a single coalescent history will not contain multiple polymorphisms). To explore this first condition, we tried decreasing the amount of simulated data in *Figure 2* from the original $10^9$ bases. MAGIC was consistently able to infer the Laplace transform (and therefore the probability distribution) when using only $10^8$ bases (i.e., approximately one human chromosome), but when using just $10^7$ bases succeeded in only some simulations (*Figure 8*, top). To explore the second condition, we re-simulated the 'bottleneck' history with increasing values of the crossover rate, $\rho$ (and without gene conversion). MAGIC accurately inferred the full Laplace transform until crossovers became more frequent than mutations, $\rho \gtrsim \mu$ (*Figure 8*, bottom), at which point it was only able to infer for small values of $z$, where the inference primarily depends on short windows (*Figure 6*, right panels).

## Implementation

The code for MAGIC is written in Python and is available at https://github.com/weissmanlab/magic (*Weissman, 2017*). It uses the same input format as MSMC and the msmc-tools suite. A copy is archived at https://github.com/elifesciences-publications/magic.

## Data processing

We use the 69 Genomes Diversity Panel from Complete Genomics (*Drmanac et al., 2010*), and use msmc-tools (*Schiffels and Durbin, 2014*) to turn the data into a list of SNPs. We split the genome into windows of 80 bp, count the number of SNPs in each window, and then repeatedly merge all windows in pairs and re-count to get the SNP count distribution at successively larger length scales (*Figure 1*, bottom left; *Figure 6*, top left). (To correct for uneven sequencing coverage across windows, all windows with <80% coverage were dropped, and all with >80% coverage were down-sampled to 80%.) This gives us SNP count distributions at a range of length scales for every chromosome of every individual in the data set.

## Simulated data

All coalescent simulations were done in ms (*Hudson, 2002*). To make the simulations computationally tractable, genomes were assembled from independently simulated 'chromosomes' of $10^7$ bases each.

For the test demographic scenarios in *Figure 2* and *Figure 3*, the per-base mutation rate was $\mu = 10^{-3}/(4N_0)$. (ms is parametrized in terms of the present population size, $2N_0$.) For the 'bottleneck' scenario, the demography was given by the command '-eG .3 10 -eG .4 –10 -eG .6 0'; for 'repeated bottlenecks', by '-eG .1 –10 -eG .3 10 -eG .5 –10 -eG .7 10 -eG .9 –10 -eG 1.1 10 ''; and for 'admixture', by '-es .1 1 .5 -ej 2 1 2'. See *Figure 9* for schematics. For the pairwise simulations, each sample consisted of 100 chromosomes with recombination rates as listed in *Figure 2*. For the larger-sample simulations in *Figure 3*, each sample consisted of 10 chromosomes with per-base rate of crossovers $\rho = \mu/5$, and per-base rate of initiation of gene conversion $g = \mu/20$ with mean tract length $\lambda = 1\text{kb}$.

## Acknowledgements

We thank Paul Marjoram, Magnus Nordborg, Stephan Schiffels, and Diethard Tautz for their very thoughtful and helpful review, with additional thanks to Dr. Schiffels for help with MSMC. We also thank Kelley Harris and Rasmus Nielsen for help with human population genetic data, Peter Ralph

and Benjamin H. Good for discussions of the mathematical analysis, Julia Palacios for comments on the preprint, Miaoyan Wang for comments on the beta version of the software, and Razib Khan for suggesting the name of the method.

## Additional information

### Funding

| Funder | Grant reference number | Author |
|---|---|---|
| Simons Foundation | Simons Investigator Award | Oskar Hallatschek |
| National Institute of General Medical Sciences | R01GM115851 | Oskar Hallatschek |

The funders had no role in study design, data collection and interpretation, or the decision to submit the work for publication.

### Author contributions
DBW, Conceptualization, Software, Investigation, Writing—original draft, Writing—review and editing; OH, Conceptualization, Investigation, Writing—original draft, Writing—review and editing

### Author ORCIDs
Daniel B Weissman, http://orcid.org/0000-0002-7799-1573

## Additional files

### Major datasets
The following previously published dataset was used:

| Author(s) | Year | Dataset title | Dataset URL | Database, license, and accessibility information |
|---|---|---|---|---|
| Drmanac R, Sparks AB, Callow MJ, Halpern AL | 2010 | 69 Genomes | http://www.completegenomics.com/public-data/69-genomes/ | Publicly available at the 69 Genomes Data website (download link: ftp://ftp2.completegenomics.com/) |

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
