## [Decision Letter]

Thank you for submitting your article "Minimal-assumption inference from population-genomic data" for consideration by *eLife*. Your article has been favorably evaluated by Diethard Tautz (Senior Editor) and three reviewers, one of whom, Magnus Nordborg (Reviewer #1), is a member of our Board of Reviewing Editors. The following individuals involved in review of your submission have agreed to reveal their identity: Stephan Schiffels (Reviewer #2); Paul Marjoram (Reviewer #3).

The reviewers have discussed the reviews with one another and the Reviewing Editor has drafted this decision to help you prepare a revised submission.

Summary:

Polymorphism data contains information about the evolutionary history of the population, and can be used for inference about the process that gave rise to the data. In the era of cheap genome sequencing, this is of great interest. However, generally speaking, the polymorphism data reflects an underlying "ancestral recombination graph", which, in itself, is a product of the evolutionary process. If we could infer this graph, we would not need the polymorphism data. This theoretical paper describes a method for extracting information about this graph under a fairly general model – information that can then be used for further dissection of the evolutionary process. As is demonstrated in the paper, this can be massively more efficient than trying to infer the details of this process directly from the polymorphism data.

Essential revisions:

The analysis of real data and its presentation could be improved, and help readers understand the method better. For example, Figure 4 shows that MAGIC apparently doesn't improve population size estimates in Yoruba over MSMC, even though the MSMC results are based on single individuals while MAGIC analyses all 9 simultaneously. At the same time, when estimating tip branch lengths, Figure 4 (right hand side) shows impressively how MAGIC's estimates are contradicting the Ne model from both MSMC and MAGIC based on pairwise coalescence times, thus perhaps revealing that the model is not good. This advantage should be made clearer, and it may also be useful to point out potential ways forward. For example, joint analysis of pairwise coalescence times and tip branch lengths might suggest better models, or generally improve estimates for certain parameter values?

This would emphasize MAGIC's utility as a flexible analysis tool that can handle large data sets.

---

## [Author Response]

*Essential revisions:*

*The analysis of real data and its presentation could be improved, and help readers understand the method better.*

For the analysis, the switch to the piecewise-exponential form has slightly improved the performance, and the additional curves in Figure 4 hopefully give a somewhat better sense for the levels and forms of noise and bias in the method.

*For example, Figure 4 shows that MAGIC apparently doesn't improve population size estimates in Yoruba over MSMC, even though the MSMC results are based on single individuals while MAGIC analyses all 9 simultaneously.*

Yes, this is an important point. We have added it in the first paragraph of the subsection “Human data”.

*At the same time, when estimating tip branch lengths, Figure 4 (right hand side) shows impressively how MAGIC's estimates are contradicting the Ne model from both MSMC and MAGIC based on pairwise coalescence times, thus perhaps revealing that the model is not good. This advantage should be made clearer, and it may also be useful to point out potential ways forward. For example, joint analysis of pairwise coalescence times and tip branch lengths might suggest better models, or generally improve estimates for certain parameter values?*

*This would emphasize MAGIC's utility as a flexible analysis tool that can handle large data sets.*

We have only added a few lines of text to this section and the Discussion, but we hope that they help make things a little clearer. The key part is –at the end of the subsection “Human data” – we think that MAGIC can be used with ad hoc ABC to match multiple inferred branch length distributions, or to check the results of multiple stand-alone inference methods. We have also added more on this point in the second paragraph of the subsection “Approach” and in the fourth paragraph of the Discussion.